# Impact of Clear Aligners versus Fixed Appliances on Periodontal Status of Patients Undergoing Orthodontic Treatment: A Systematic Review of Systematic Reviews

**DOI:** 10.3390/healthcare11091340

**Published:** 2023-05-06

**Authors:** Federica Di Spirito, Francesco D’Ambrosio, Davide Cannatà, Vincenzo D’Antò, Francesco Giordano, Stefano Martina

**Affiliations:** 1Department of Medicine, Surgery and Dentistry “Scuola Medica Salernitana”, University of Salerno, Via S. Allende, 84081 Baronissi, Italy; francesco87fd@libero.it (F.D.); davide2897@icloud.com (D.C.); smartina@unisa.it (S.M.); 2School of Orthodontics, Department of Neurosciences, Reproductive Sciences and Oral Sciences, University of Naples Federico II, 80131 Naples, Italy

**Keywords:** periodontal health, orthodontic treatment, fixed appliances, clear aligners, oral hygiene, biofilm control, gingivitis

## Abstract

The present umbrella review of four systematic reviews with meta-analysis aimed to assess whether clear aligners are associated with better periodontal conditions compared with fixed appliances in patients undergoing orthodontic treatment. The present study protocol was developed in accordance with the PRISMA statement before the literature search, data extraction, and analysis and was registered on PROSPERO (CRD42023401808). The question formulation, search, and study selection strategies were developed according to the PICO model. Systematic reviews with a meta-analysis published in English without date restriction were electronically searched across the Cochrane Library, Web of Science (Core Collection), Scopus, EMBASE, and MEDLINE/PubMed databases until 10 February 2023. An assessment of study quality was performed using the AMSTAR 2 tool. Differences in the PI, GI, and BOP in the short- and medium-term follow-ups, in the PPD in long-term follow-up, and the gingival recessions in the short-term follow-up were found between subjects with clear aligners and fixed appliances, revealing a slight tendency for clear aligners to be associated with healthier periodontal conditions. However, even if statistically significant, such differences would be negligible in a clinical environment. Therefore, the impact of orthodontic treatment with clear aligners and fixed appliances on periodontal health status should be considered comparable.

## 1. Introduction

Periodontal health is defined by the absence of microscopically and macroscopically detectable signs of inflammation that may affect periodontal physiology [1].

Periodontal health maintenance in patients undergoing orthodontic treatment depends on several factors, including the patient’s oral hygiene habits and biofilm control, periodontal host–microbe homeostasis, periodontal phenotype, especially with regard to buccal bone plate width, systemic conditions and diseases (e.g., diabetes mellitus) directly and indirectly affecting the periodontal status and oral microbiome, and personal habits (e.g., smoking) [2,3].

Traditional orthodontic treatment encourages tooth movement to correct dental malocclusion through appliances fixed to teeth surfaces, such as orthodontic bands and brackets, archwires, ligatures, and auxiliaries [4,5].

Fixed orthodontic appliances often complicate oral hygiene procedures [6] and facilitate plaque accumulation on both teeth and the appliances’ surfaces [7,8,9]. Indeed, the biofilm control and clinical periodontal inflammatory parameters are generally worse in patients undergoing orthodontic treatment with fixed appliances than in non-orthodontic patients [9]. Hence, a combination of manual, orthodontic, or powered brushing, motivational aids, and organic products, or the short-term use of chlorhexidine mouthwashes could be recommended for biofilm control and gingival inflammation reduction in subjects with fixed orthodontic treatment [10].

Clear aligners were introduced in 1999 to overcome some limitations of fixed appliances and to satisfy the esthetic and comfort requirements of patients. Indeed, orthodontic treatment with clear aligners is based on removable thermoplastic splints covering the entire dental arch, progressively moving the teeth into an ideal position [11]. The current literature describes orthodontic treatment with clear aligners as safe, comfortable, and aesthetic [12,13]. Moreover, clear aligners have been reported to offer an advantage over fixed appliances in the segmented movement of teeth and shortened treatment duration [14]. In contrast, fixed appliances seemed more effective in producing adequate occlusal contacts and controlling teeth torque and rotation [15]. Nevertheless, recent technological developments have made it possible to treat many complex malocclusions with clear aligners [16].

In addition, clear aligners can be easily removed during meals and oral hygiene procedures, allowing patients to effectively control gingival biofilm [2,13] and, thus, presumably maintain healthier periodontal conditions during orthodontic treatment as compared to traditional fixed appliances. This is particularly important for adults, who have a higher prevalence of periodontitis, seeking orthodontic treatment [17].

However, the evidence on the periodontal health status of subjects undergoing orthodontic treatment with clear aligners and fixed appliances remains contradictory [18].

Hence, the aim of the present umbrella review was to summarize the current evidence in order to assess whether clear aligners are associated with a more beneficial impact on periodontal health status compared to fixed appliances in patients undergoing orthodontic treatment.

The null hypothesis was that the impact of orthodontic treatment with clear aligners and fixed appliances on periodontal health is comparable.

## 2. Materials and Methods

### 2.1. Study Protocol

The present study protocol, registered in the PROSPERO International Prospective Register of Systematic Reviews (CRD42023401808), was developed before the literature search, data extraction, and analysis and performed in accordance with the PRISMA (Preferred Reporting Items for Systematic Reviews and Meta-analyses) statement (Appendix A) [19,20].

The study question definition, search strategies, and study selection criteria were developed according to the PICO model [21]. The study question was “Are clear aligners more beneficial for a healthy periodontal status than fixed appliances?” and focused on:

P—Population: patients undergoing orthodontic treatment (with no age or gender restriction) with fixed orthodontic appliances or clear aligners;

I—Intervention: orthodontic treatment with clear aligners (any type);

C—Comparison: orthodontic treatment with fixed (vestibular or lingual) appliances;

O—Outcome(s): periodontal health status as measured by clinical indices (excluding self-report).

### 2.2. Search Strategy

An electronic literature search for systematic reviews with meta-analysis, published in English, without date restriction, and related to the periodontal health status of patients undergoing orthodontic treatment was independently conducted by two independent reviewers (D.C. and F.D.A.), through February 2023, across the Cochrane Library, Web of Science (Core Collection), Scopus, EMBASE, and MEDLINE/PubMed databases. The search strategy was performed according to medical subject heading terms (Mesh), if any, and non-mesh terms; the mesh and non-mesh terms were also combined with Boolean operators, as shown in Table 1.

The following filters were applied: “Review (English)” and “Topic” in the Web of Science database; “Review (English) in the Scopus database; “Systematic Review (English)” and “Meta-analysis (English)” in the MEDLINE/PubMed database; “Systematic Review (English)” and “Meta-analysis (English)” in the EMBASE database; “Review” and “Title, Abstract, Keyword” in the Cochrane Library.

Moreover, an exploration of the grey literature (unpublished studies) was performed in the OpenGrey database.

### 2.3. Study Selection and Eligibility Criteria

Collected citations were recorded, and duplicates were eliminated using the reference management tool EndNoteTM (Clarivate). Two reviewers (D.C. and F.D.A.) independently screened the remaining records and identified potentially relevant titles and abstracts eligible for further analysis.

The full texts of those records that met the eligibility criteria and the ambiguous title/abstracts were obtained without requiring contact with the study authors. The two authors (D.C. and F.D.A.) reviewed the full texts independently. The level of agreement between authors in the study selection process was assessed through the Cohen-weighted kappa (κ) coefficient, considering a minimum threshold value of 0.61 (substantial) [22]. The opinion of a third author (F.D.S.) was sought, if necessary, in case of disagreement.

The inclusion and exclusion criteria are reported in Table 2.

### 2.4. Data Extraction and Collection

Data were extracted independently by two authors (D.C. and F.D.A.). A dedicated data extraction form was used which was developed before the start of the study and followed the models proposed for the intervention reviews of RCT and non-RCTs [23]; a third author (F.D.S.) was consulted when necessary. The Cohen kappa coefficient was used to assess the inter-examiner reliability in the data extraction and collection process [24].

For each systematic review included in the present review, the following data were collected:-The first author, year, journal, and funding;-The design and number of studies included in the qualitative syntheses of each review;-The number of studies included in the quantitative meta-analysis;-The population characteristics: sample size, age range, and gender ratio;-The intervention group data: treatment performed (type, characteristics, duration, and follow-up of the orthodontic treatment with clear aligners);-The control group data: orthodontic treatment performed (type, characteristics, duration, and follow-up of the orthodontic treatment with fixed appliances);-The statistically significant periodontal outcomes (clinical parameters, such as the plaque index “PI”, gingival index “GI”, periodontal probing depth “PPD”, gingival recession, and others; crevicular parameters; any other parameters to assess the periodontal health status or gingival biofilm accumulation reported in the systematic reviews with a meta-analysis included);-The conclusion(s) of the study.

### 2.5. Data Synthesis

A narrative synthesis was performed, focusing on the population studied, the intervention, and periodontal outcomes. Microsoft Excel 2016 software (Microsoft Corporation, Redmond, WA, USA) was used to qualitatively summarize the data from the included studies in descriptive statistical analysis and evaluate the clinical periodontal outcomes concerning the orthodontic appliances used (fixed vs. clear aligners).

### 2.6. Quality Assessment

Two authors (D.C. and F.D.A.) assessed the included reviews independently, using the Assessing the Methodological Quality of Systematic Reviews (AMSTAR) 2 tool composed of 16 items [25]. Any disagreement was initially resolved by discussion or in conjunction with a third author (S.M.) if necessary.

## 3. Results

### 3.1. Study Selection

The electronic search of all databases yielded 166 references. After duplicate elimination, 134 references remained. In reviewing the titles of the 134 entries, 121 were excluded because the subject was not relevant or because the type of the article was not a systematic review.

Abstracts of the remaining 13 articles were obtained, and 6 did not meet the eligibility criteria and were therefore excluded.

Full texts of the remaining seven articles were screened. Contacting the authors to obtain the full text or further information was unnecessary. Three articles were additionally excluded because they did not meet the inclusion criteria for this study. Specifically, two systematic reviews synthesized findings from studies comparing different types of orthodontic aligners [2,26] and one study did not provide a quantitative meta-analysis [27] (Table 3).

Finally, four articles [28,29,30,31] from the electronic search were included in this umbrella review. No additional studies that were compatible with the eligibility criteria were found by screening grey literature or manually reviewing the reference lists of the included articles. The level of agreement between the examiners in the selection process (Cohen kappa coefficient) was 0.79 (substantial).

Figure 1 shows the flow diagram from the study selection, which included electronic searching databases and registers.

### 3.2. Study Characteristics

None of the four systematic reviews with a meta-analysis [24,25,26,27] included in this umbrella review reported being externally funded [28,29,30,31].

Two systematic reviews [29,30] provided qualitative and quantitative data synthesis from all their included studies. The quantitative meta-analysis of the other two systematic reviews [28,31] included only a subset of the reviewed studies.

The total sample size was 2042 orthodontic patients, including 681 males and 850 females, aged 10 to 51 years, although gender and age were not reported for 551 and 464 subjects, respectively.

Participants were grouped as follows: 978 fell into the intervention group treated with clear aligners and 1064 comprised the control group treated with fixed vestibular or lingual appliances. Specifically, of the 978 subjects in the intervention group, 411 were treated with Invisalign^®^, 15 with PET-G (polyethylene terephthalate glycol) aligners, 20 with AirNivol aligners, and the remaining 532 participants with unspecified types of aligners. Of the 1064 subjects in the control group, 30 underwent orthodontic treatment with a fixed lingual appliance; specifically, 461 with a fixed buccal appliance; 35 with elastomeric ligated brackets; and 15 with self-ligating fixed appliances; no further information on fixed appliances was provided for the remaining 523 participants.

None of the included systematic reviews reported the total orthodontic treatment duration for clear aligners and fixed appliances. Follow-up for both the intervention and control groups ranged from 1 month to more than two years.

In all included reviews, clinical periodontal parameters were assessed. Specifically, the plaque index (PI), gingival index (GI), and periodontal probing depth (PPD) were measured in all four included studies [28,29,30,31]. One systematic review recorded the sulcus bleeding index (SBI) [29,31] and another systematic review reported gingival recessions [31]. Any other parameter (e.g., radiographic, crevicular) assessing periodontal health status was registered.

A subgroup analysis of data according to study type (RCTs and non-randomized studies) was performed only in one study [28], while a subgroup analysis based on follow-up time was performed in all four meta-analyses [28,29,30,31].

The inter-examiner reliability (Cohen kappa coefficient) in the data extraction and collection process was 0.87 (almost perfect).

The extracted data are reported in Table 4.

### 3.3. Quality of the Included Systematic Reviews

According to the AMSTAR 2 checklist, the quality of the included reviews was variable: two [29,30] were classified as being of low-quality evidence, one [28] as being of moderate-quality evidence, and one [31] as being of high-quality evidence. Most of the AMSTAR 2 items were covered to varying degrees.

The two systematic reviews considered low-quality evidence [29,30] had the same critical deficiency: the authors had not provided a list of excluded studies and had not justified the reasons for exclusion. There was complete agreement among reviewers on the quality assessment.

The quality assessment and level of evidence of the systematic reviews included in the present study according to the AMSTAR 2 tool are shown in Table 5 and Table 6.

### 3.4. Synthesis Results for Periodontal Clinical Parameters

The five periodontal clinical parameters recorded in the systematic reviews included the plaque index (PI), gingival index (GI), sulcus bleeding index (SBI), periodontal probing depth (PPD), and gingival recessions. No data concerning the clinical attachment level (CAL) were retrieved from the systematic reviews currently considered.

#### 3.4.1. Plaque Index

All four systematic reviews included in the present umbrella review [28,29,30,31] examined the PI in groups treated with clear aligners and fixed orthodontic appliances.

A significant difference in the PI between subjects undergoing orthodontic treatment with clear aligners and fixed appliances was reported in the short-term (from baseline to 2–3 months) follow-up (low to moderate evidence), ranging from −0.35 to −0.69 [28,29,30], and the medium-term (from baseline to 6–9 months) follow-up (low to high evidence), ranging from −0.91 to −1.10 [29,30,31].

At the long-term follow-up (from baseline to 12 months or more), no difference in the PI was found between treatment with clear aligners and fixed appliances [28,31].

#### 3.4.2. Gingival Index

All four meta-analyses included in this review assessed differences between the GI in participants undergoing orthodontic treatment with clear aligners and fixed appliances [28,29,30,31].

A significant difference in the GI between the intervention group (clear aligners) and the control group (fixed appliances) was reported at the short-term (from baseline to 2–3 months) follow-up (moderate evidence) (at 1 month: MD, −0.24, 95% CI, −0.35 to −0.12; at 3 months MD, −0.63, 95% CI, −1.22 to −0.04) [28] and in the medium-term follow-up (from baseline to 6–9 months) (low evidence) (MD, −0.14, 95% CI, −1.95 to −0.34) [30]. No difference was found in the long-term follow-up (from baseline to 12 months or longer) (high evidence) [31].

#### 3.4.3. Periodontal Probing Depth

All systematic reviews included evaluated the PPD values in clear aligners vs. fixed orthodontic appliance groups [28,29,30,31].

Some findings, albeit of low evidence, pointed out better PPD values in orthodontic patients treated with clear aligners compared to those who underwent fixed orthodontic treatment in the short-term follow-up (from baseline to 2–3 months) (at 3 months: MD, −0.26, 95% CI, −0.52 to −0.01 [30] and in the medium-term (from baseline to 6–9 months) follow-up (at 6 months: MD, −0.42, 95% CI, −0.83 to −0.01) [30].

Evidence ranging from moderate to high supported the role of clear aligners in reducing the worsening of PPD values during orthodontic treatment in the long-term follow-up (from baseline to 12 months or more), with a mean difference between the intervention and control groups of −0.45 to −0.9 [28,31].

#### 3.4.4. Sulcus Bleeding Index

One meta-analysis [26] included in this review compared the SBI values in subjects undergoing orthodontic treatment with clear aligners and fixed appliances and revealed that clear aligners, compared with fixed orthodontic appliances, had a significantly lower SBI status in the short-term (after 1 month: SMD, −0.44; 95% CI: −0.70 to −0.19; after 3 months: SMD, −0.49, 95% CI, −0.93 to −0.05) and medium-term (after 6 months: SMD, −0.91, 95% CI, −1.47 to −0.35) follow-ups.

#### 3.4.5. Gingival Recession

Gingival recession was reported in only one of the systematic reviews presently considered [31], but no meta-analysis was performed because this outcome was recorded in only one study included in the systematic review.

In patients who underwent orthodontic treatment with fixed appliances, the gingival recession was statistically significantly higher at the 3-month follow-up (−0.85 mm ± 0.45) than at baseline (−0.67 mm ± 0.51). In contrast, no differences in the position of the gingival margin between the baseline and follow-up were observed in the group of subjects with clear aligners.

Table 7a–d shows a summary of the periodontal outcomes used to assess the periodontal health status of patients undergoing orthodontic treatment with clear aligners and fixed appliances, based on the timing of the follow-up examination.

## 4. Discussion

Periodontal complications have been reported to be one of the most common side effects associated with orthodontic treatment [32].

In periodontally healthy orthodontic patients, gingivitis is the most critical periodontal complication and rarely progresses to periodontal disruption during orthodontic treatment [33].

However, periodontitis prevalence increases with age [17]. Consequently, as more adults have sought orthodontic treatment in recent decades, the number of periodontitis patients undergoing orthodontic treatment has increased significantly [34]. In these patients, the periodontal complications of orthodontic treatment may include additional attachment loss and the progression or recurrence of periodontitis [35]. Indeed, orthodontic forces may increase periodontal tissue and especially bone loss at periodontitis sites by further upregulating Interleukin (IL)-6, which is already elevated by bacterially induced periodontal inflammation [36].

Recent studies suggest that clear aligners may be the first treatment option in patients at risk for gingivitis or periodontitis, favoring the maintenance of better periodontal health conditions in patients at risk for gingivitis or periodontitis, especially adults [37,38,39,40].

Considering the importance of limiting periodontal complications during orthodontic treatment, the present umbrella review of systematic reviews and meta-analyses aimed to summarize the current evidence on the impact of clear aligners on periodontal health status compared to fixed appliances in patients undergoing orthodontic treatment.

A total of four studies [28,29,30,31] were included, with a sample of 2042 participants aged 10 to 51 years undergoing orthodontic treatment with clear aligners or fixed appliances. This study population is representative of the sociodemographic characteristics of the current population of orthodontic patients [41]. Indeed, currently, both young people and adults seek orthodontic treatment. Moreover, in all age groups of patients undergoing orthodontic treatment, there is a higher prevalence of female patients, which is consistent with the sample of the present study that has an M:F ratio of 1:1.3 [41].

None of the studies reported the total orthodontic treatment duration with clear aligners or fixed appliances. However, this does not seem relevant, as soft tissue inflammation can develop rapidly within the first few months of treatment, depending more on individual susceptibility than treatment duration [42].

Participant follow-up ranged from 1 month to more than 2 years, including the assessment of periodontal outcomes at a short-term follow-up (from baseline to 2–3 months), medium-term follow-up (from baseline to 6–9 months), and long-term follow-up (from baseline to 12 months or more). The latter approximates the end time of orthodontic treatment, as the mean duration is 19.9 months (MD, 19.9, 95% CI, 19.58 to 20.22 months).

### 4.1. Impact of Clear Aligners versus Fixed Appliances on Periodontal Conditions

Investigators in the included studies used five clinical indices to assess periodontal health status: the plaque index (PI), gingival index (GI), sulcus bleeding index (SBI), periodontal probing depth (PPD), and gingival recession.

The plaque index (PI) is a commonly used clinical index to assess oral hygiene status based on the accumulation of gingival biofilm around the teeth, gingiva, and gingival sulcus or periodontal pockets [43]. Gingival biofilm is a polymicrobial biofilm composed of diverse bacterial complexes that mutually benefit from coaggregation, adhesion, and metabolic interactions [44,45] and is the primary etiologic factor in the development, progression, and recurrences of periodontal inflammation [46,47,48]. The main biofilm-induced periodontal diseases are gingivitis, a non-destructive inflammation of the gingiva that is reversible by controlling the biofilm, and periodontitis, which instead leads to the irreversible loss of attachment, periodontal ligament, and alveolar bone [49,50,51] and is also dependent on the individual’s susceptibility and responsiveness to the inflammatory insult [52,53,54].

Orthodontic patients always have significantly worse PI scores, indicating a worse oral hygiene status compared to non-treated individuals [55]. In turn, there is a good consensus in the literature, confirmed by the present results, that clear aligners provide significantly better control of biofilm accumulation than fixed orthodontic appliances, especially during the first year of treatment. Indeed, differences in the PI between patients with clear aligners and fixed appliances were reported at short-term (from −0.35 to −0.69) and medium-term follow-ups (−0.91 to −1.10) (Table 7a). These findings may be explained by the evidence that orthodontic brackets, bands, and archwires promote biofilm accumulation, retain more plaque, and hinder its effective removal [46]. In addition, it may be speculated that after the first year of fixed orthodontic treatment, on average, patients with fixed appliances become more adept at oral hygiene. On the other hand, clear aligners cover at least most of the crown, prevent biofilm accumulation [27], and can be removed, allowing patients to perform their oral hygiene procedures under optimal conditions [27].

The clinical evaluation of gingival inflammation was recorded through the gingival index (GI) and sulcus bleeding index (SBI), measuring gingival edema and erythema and assessing the presence and severity of periodontal bleeding [43,56].

Because orthodontic fixed appliances generally reduce the effectiveness of biofilm control in patients, the associated risk of local gingivitis occurrence is expected to increase [46,57]. Furthermore, biofilm accumulation may favor subgingival periodontal pathogens and induce the release of periodontal proinflammatory cytokines, which in turn leads to periodontal tissue destruction [58,59]. Conversely, clear aligners, which prevent biofilm accumulation, may potentially limit the risk of inflammation and subsequent destruction of periodontal tissue during orthodontic treatment [27].

Accordingly, the included meta-analyses’ results reported evidence of the significantly better control of gingival inflammation among patients with removable rather than fixed appliances. Indeed, differences in the GI were reported in the short-term and medium-term follow-ups (at 1 month: MD, −0.24, 95% CI, −0.35 to −0.12; at 3 months MD, −0.63, 95% CI, −1.22 to −0.04; at 6 months: MD, −0.14, 95% CI, −1.95 to −0.34) (Table 7b). Similarly, differences in the SBI were found in the short-term (at 1 month: SMD, −0.44; 95% CI: −0.70 to −0.19; at 3 months: SMD, −0.49, 95% CI, −0.93 to −0.05) and mid-term (at 6 months: SMD, −0.91, 95% CI, −1.47 to −0.35) follow-ups (Table 7d).

Recent studies have found that the worsening of periodontal probing depth values (PPD) during orthodontic treatment is mainly due to the bacterial biofilm-induced inflammation of the gingiva, which can lead to gingival overgrowth and periodontal pseudopockets [60].

Retrieved findings supported that clear aligners are associated with better PPD values at short- and medium-term follow-ups (after 3 months: MD, −0.26, 95% CI, −0.52 to −0.01; after 6 months: MD, −0.42, 95% CI, −0.83 to −0.01) (Table 7c). In addition, moderate to high evidence supported the role of clear aligners in limiting the worsening of PPD values during orthodontic treatment at the long-term follow-up (from −0.45 to −0.93) (Table 7c) compared with fixed orthodontic appliances.

Some authors suggested that orthodontic treatment could be associated with gingival recession and loss of clinical attachment level [61]. Crego-Ruiz et al. [31] reported a significant increase in gingival recession at the 3-month follow-up in patients with fixed orthodontic appliances but not in subjects treated with clear aligners. However, no data concerning the clinical attachment level (CAL) were retrieved from the systematic reviews currently considered.

### 4.2. Clinical Considerations and Implications

Almost all periodontal indices showed a slight tendency for clear aligners to be associated with healthier periodontal conditions by limiting gingival biofilm accumulation and periodontal inflammation. However, some clinical considerations should be made.

Although statistically significant, the difference in PPD between subjects with clear aligners and patients undergoing fixed orthodontic treatment was less than one millimeter (from −0.45 to −0.93) (Table 7c), which would be practically negligible in a clinical setting [62,63,64].

In addition, when considering the clinical relevance of the differences in the PI, GI, and SBI between patients with clear aligners and patients with fixed appliances, it is important to remember that these indices are ordinal variables. Therefore, if the mean difference between subjects with clear aligners and those with fixed appliances is between 0 and 1, both approaches would likely receive the same index value because the difference is less than a whole point. Consequently, the differences in periodontal indices between the two treatment groups should be considered clinically significant only if they are greater than 1. Accordingly, only the difference in the PI score reported during the medium-term follow-up can assume clinical relevance.

Regarding gingival recessions, although clear aligners seemed to be associated with a more stable gingival margin position, this outcome was recorded in a short-term follow-up and only one study, and thus, it needs to be confirmed by further studies with a long-term follow-up.

Based on the above, the current evidence remains insufficient to determine whether clear aligners are associated with healthier periodontal conditions. In addition, no evidence suggested that clear aligners should be a first-line treatment option in patients, especially adult ones, at risk for gingivitis or periodontitis.

Accordingly, rather than the choice of treatment modality, the establishment of an appropriate periodontal surveillance and health promotion program for the adequate control of periodontal biofilm and inflammation should be considered as an effective preventative measure for periodontal complications during orthodontic treatment [10].

Furthermore, since uncontrolled periodontal inflammation during orthodontic treatment is known to promote the progression of periodontitis and tissue destruction [65,66,67], a comprehensive diagnosis that takes into account not only the patient’s orthodontic problems but also their periodontal needs to achieve and maintain periodontal health is strongly recommended before starting orthodontic treatment [68].

When interpreting the clinical implications of the results, some limitations of the present systematic review of systematic reviews should be considered.

First, several studies were included in more than one of the systematic reviews currently included. Second, high heterogeneity was found in almost all the meta-analyses performed in the included systematic reviews. This could be due to differences in patients’ periodontal self-care and home care instructions given by physicians. In addition, heterogeneous data, especially regarding the timing of follow-up, and lack of data on the characteristics and duration of orthodontic treatment precluded the possibility of performing a meta-analysis.

Further studies investigating the clinical relevance of the differences between the impact of clear aligners and fixed appliances on periodontal health status are needed for definitive conclusions.

## 5. Conclusions

The present study included four systematic reviews examining the impact of clear aligners compared with fixed appliances on the periodontal health status of patients undergoing orthodontic treatment.

Clear aligners provided a significantly better control of biofilm accumulation than fixed orthodontic appliances, especially during the first year of treatment (PI: MD from −0.35 to −1.10); however, no differences were found during the long-term follow-up. Similarly, the gingival inflammatory status was significantly better controlled in patients with removable rather than fixed appliances at short- and medium-term follow-ups (GI at 1 month: MD, −0.24, 95% CI, −0.35 to −0.12; at 3 months MD, −0.63, 95% CI, −1.22 to −0.04; at 6 months: MD, −0.14, 95% CI, −1.95 to −0.34). In addition, there was moderate to strong evidence that clear aligners limited the worsening of PPD values during orthodontic treatment at the long-term follow-up (from −0.45 to −0.93).

However, such differences in periodontal outcomes between subjects with clear aligners and fixed appliances were statistically significant, but practically negligible in the clinical context. Indeed, differences in the GI and SBI (ordinal variables) were less than 1, and differences in the PPD were less than 1 mm (hardly measurable).

Regarding gingival recession, there is not enough evidence supporting that clear aligners might increase or decrease gingival recessions compared to fixed appliances.

Given the current state of knowledge, the impact of orthodontic treatment with clear aligners and fixed appliances on periodontal health should be considered comparable, and there is no evidence to support the choice of clear aligners as the first treatment option in patients at risk for gingivitis or periodontitis.

## Figures and Tables

**Figure 1 healthcare-11-01340-f001:**
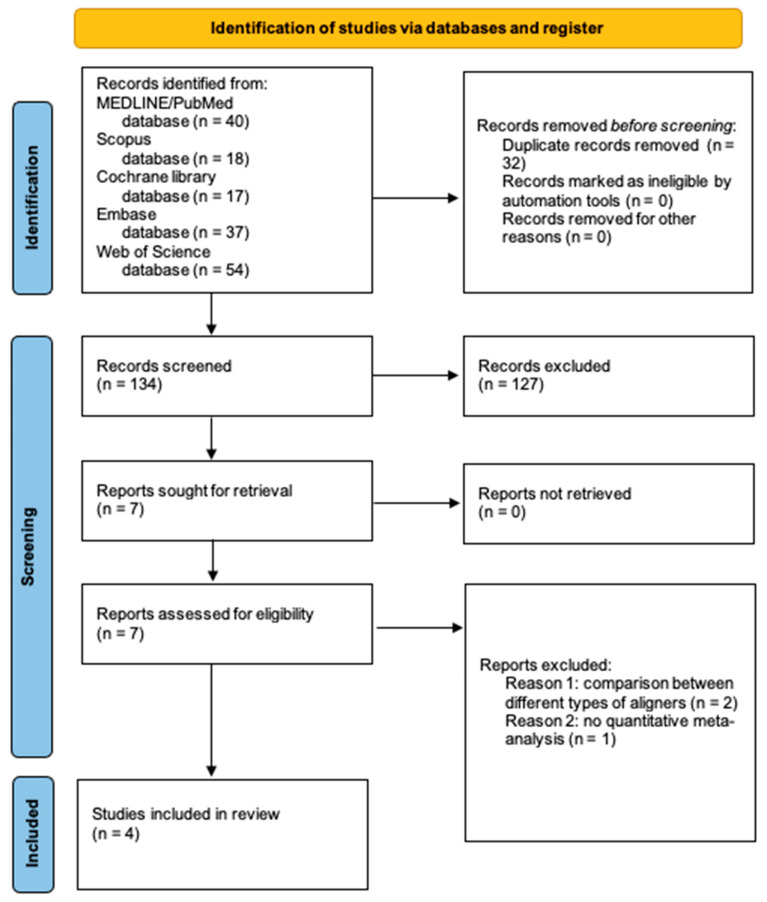
PRISMA 2020 flow diagram for selecting new systematic reviews, including searches of databases and registers only.

**Table 1 healthcare-11-01340-t001:** Search strategy.

DATABASES	Cochrane Library, Web of Science, Scopus, EMBASE, and MEDLINE/PubMed
DATE	15/02/2023
STRATEGY	#1 AND #2
#1	“periodontal health” OR “gingival health” OR “oral hygiene” OR “periodontal indices” OR “bacteria” OR “periodontopathogen*” OR “pathogen” OR “microorganism” OR “microbe” OR “plaque” OR “biofilm” OR “microflora” OR “microbiome”
#2	“orthodontic appliances” OR “fixed appliances” OR “brackets” OR “aligner” OR “Invisalign” OR “clear aligner”

**Table 2 healthcare-11-01340-t002:** Inclusion and exclusion criteria.

**Inclusion Criteria**	Exclusion Criteria
Study design: systematic reviews with a meta-analysis Population: patients undergoing orthodontic treatment (with no age or gender restriction) Intervention: clear aligners (any type) Comparison: fixed orthodontic appliances (any type) Language: English	Study design: narrative review; systematic review of systematic review; systematic review without a meta-analysis Systematic review comparing different types of orthodontic aligners Duplicate records No relevant title or abstract

**Table 3 healthcare-11-01340-t003:** Studies excluded and the reasons for their exclusion.

Authors, Year	Reason for Exclusion
Oikonomou et al., 2021 [26]	The studies included in the systematic review also compared different types of orthodontic aligners
Rossini et al., 2015 [2]	The studies included in the systematic review also compared different types of orthodontic aligners
Partouche et al., 2022 [27]	No quantitative meta-analysis

**Table 4 healthcare-11-01340-t004:** Data collected from the studies included in the present systematic review of systematic reviews: general information: First author, year, journal of publication, reference number, and funding; methods: study characteristics (number and design of studies included in the qualitative synthesis; number of studies included in the qualitative meta-analysis), participants (sample size, age, and gender), intervention (type, characteristics, duration, and follow-up), and comparison (type, characteristics, duration, and follow-up); periodontal outcomes; and conclusion(s).

Included Systematic Review	Studies Included in Qualitative and Quantitative Synthesis	Sample	Periodontal Outcomes Statistically Significant (*p* < 0.05)	Conclusion(s)
**Jiang et al., 2018** [28] **J Am Dent Assoc.** **No funding**	**Qualitative synthesis:** n.10 RCT (n.3) Non-RCT (n.7) **Quantitative synthesis:** n.9	**Population:** Sample size: (n.464) Age range: NDF Male/female ratio: NDF **Intervention group** (n.207): Type: clear aligners Characteristics: NS Duration: NDF Follow-up: from 1 to 31.6 mo. **Control group** (n.257): Type: fixed appliances Characteristics: fixed lingual orthodontic appliance (n.30); fixed buccal orthodontic appliance (n.227) Duration: NDF Follow-up: from 1.0 to 31.6 mo.	**PI:** *Subgroup analyses according to study type* RCTs (n.67): NSS Non-RCT (282): MD, −0.21; 95% CI, −0.45 to 0.03; *p* < 0.001 Subtotal (n.349): MD, −0.53; CI 95%, −0.85 to −0.20; *p* = 0.001 *Subgroup analyses according to follow-up time* 1 mo. (n.190): MD, −0.35, 95% CI, −0.57 to −0.14; *p* = 0.001 3 mo. (n.127): MD, −0.63, 95% CI, −1.22 to −0.04; *p* = 0.04 6 mo. (n.89): NSS 12 mo. (n.89): NSS Overall (n.585): MD, −0.75, 95% CI, −1.06 to −0.45; *p* < 0.001 **GI:** *Subgroup analyses according to study type* Non-RCT (360): MD, −0.27; 95% CI, −0.37 to −0.17; *p* < 0.001 *Subgroup analyses according to follow-up time* 1 mo. (n.170): MD, −0.24, 95% CI, −0.35 to −0.12; *p* = 0.001 3 mo. (n.120): MD, −0.37, 95% CI, −0.65 to −0.10; *p* = 0.007 Overall (n.290): MD, −0.30, 95% CI, −0.43 to −0.18; *p* < 0.001 **PPD:** *Subgroup analyses according to study type* RCTs (n.67): NSS Non-RCT (240): MD, −0.39; 95% CI, −0.75 to −0.03; *p* = 0.03 Subtotal (n.307): MD, −0.35; CI 95%, −0.67 to −0.03; *p* = 0.03 *Subgroup analyses according to follow-up time* 1 mo. (n.140): NSS 3 mo. (n.217): NSS 6 mo. (n.89): NSS 12 mo. (n.89): MD, −0.45, 95% CI, −0.41 to −0.10; *p* = 0.001 Overall (n.585): MD, −0.75, 95% CI, −1.06 to −0.45; *p* < 0.001	Clear aligners, in comparison with fixed appliances, allowed for significantly better periodontal health, including PI, GI, and PPD.
**Lu et al., 2018** [29] **Medicine (Baltimore)** **No funding**	**Qualitative:** n.7 Non-RCT (n.7) **Quantitative:** n.7	**Population:** Sample size: (n.368) Age range: from 15 to 40 y Male/female ratio: 126M/242F **Intervention group** (n.183): Type: clear aligners Characteristics: Invisalign^®^ (n.183) Duration: NDF Follow-up: 6 mo. **Control group** (n.185): Type: fixed appliances Characteristics: NS Duration: NDF Follow-up: 6 mo.	**PI** *Subgroup analyses according to follow-up time* 1 mo.: SMD, −0.53; 95% CI: −0.89 to −0.18; *p* < 0.05 3 mo.: SMD, −0.69, 95% CI, −1.12 to −0.27; *p* < 0.05 6 mo.: SMD, −0.91, 95% CI, −1.47 to −0.35; *p* < 0.05 Overall: SMD, −0.74, 95% CI, −1.02 to −0.46; *p* < 0.05 **GI:** *Subgroup analyses according to follow-up time* 1 mo.: NSS 3 mo.: NSS 6 mo.: NSS Overall: NSS **PPD:** *Subgroup analyses according to follow-up time* 1 mo.: NSS 3 mo.: NSS 6 mo.: NSS Overall: NSS **SBI:** *Subgroup analyses according to follow-up time* 1 mo.: SMD, −0.44; 95% CI: −0.70 to −0.19; *p* < 0.05 3 mo.: SMD, −0.49, 95% CI, −0.93 to −0.05; *p* < 0.05 6 mo.: SMD, −0.91, 95% CI, −1.47 to −0.35; *p* < 0.05 Overall: SMD, −0.40, 95% CI, −0.73 to −0.07; *p* < 0.05	Clear aligners, in comparison with fixed orthodontic appliances, allowed for a significantly lower PBI and PI indices over the course of treatment; however, no difference was found in the GI and PPD indices.
**Wu et al., 2022** [31] **Medicine (Baltimore)** **No funding**	**Qualitative:** n.13 RCT (n.13) **Quantitative:** n.13	**Population:** Sample size: (n.598) Age range: between 15.2 and 31.9 y Male/female ratio: 336M/262F **Intervention group** (n.297): Type: clear aligners Characteristics: NS Duration: NDF Follow-up: 6 mo. **Control group** (n.310): Type: fixed appliances Characteristics: NS Duration: NDF Follow-up: 6 mo.	**PI:** *Subgroup analyses according to follow-up time* 3 mo. (n.281): MD, −0.57, 95% CI, −0.98 to −0.16; *p* = 0.006 6 mo. (n.363): MD, −1.10, 95% CI, −1.60 to −0.61; *p* = 0.000 **GI:** *Subgroup analyses according to follow-up time* 3 mo. (n.175): NSS 6 mo. (n.397): MD, −0.14, 95% CI, −1.95 to −0.34; *p* = 0.005 **PPD:** *Subgroup analyses according to follow-up time* 3 mo. (n.241): MD, −0.26, 95% CI, −0.52 to −0.01; *p* = 0.047 6 mo. (n.343): MD, −0.42, 95% CI, −0.83 to −0.01; *p* = 0.045	Clear aligners are more beneficial for a healthy periodontal status since the GI, PI, and PPD indices were significantly reduced with clear aligners compared with conventional fixed orthodontic devices.
**Crego-Ruiz et al., 2023** [31] **Med Oral Patol Oral Cir Bucal.** **No funding**	**Qualitative:** n.12 RCT (n.3) Non-RCT (n.9) **Quantitative:** n.8	**Population:** Sample size: (n.612) Age range: between 10 and 51 y Male/female ratio: 219M/346F/47NDF **Intervention group** (n.291): Type: clear aligners Characteristics: Invisalign^®^ (n.228); PET-G aligners (n.15); Air-Nivol S.r.l (n.20); and NS(n.28) Duration: NDF Follow-up: from 2 to 18 mo. **Control group** (n.321): Type: fixed appliances Characteristics: fixed buccal orthodontic appliance (n.234); elastomeric ligated brackets (n.35); self-ligating fixed appliances (n.15); and NS (n.37) Duration: NDF Follow-up: from 2 to 18 mo.	**PI:** *Subgroup analyses according to follow-up time* Short-term (2–3 mo.) (n.187): NSS Mid-term (6–9 mo.) (n.203): MD, −0.99, 95% CI, −1.94 to −0.03; *p* = 0.04 Long-term (12 mo. or more) (n.108): NSS **GI:** *Subgrous analyses according to follow-up time* Short-term (2–3 mo.) (n.60): NSS Mid-term (6–9 mo.) (n.96): NSS Long-term (12 mo. or more) (n.161): NSS **PPD:** *Subgroup analyses according to follow-up time* Short-term (2–3 mo.) (n.214): NSS Mid-term (6–9 mo.) (n.82): NSS Long-term (12 mo. or more) (n.47): MD, −0.93, 95% CI, −1.16 to −0.70; *p* < 0.001 **Gingival recessions:** *3 mo.follow-up:* FA group: SS CA group: NSS	Clear aligners seem to maintain slightly better periodontal health indices. Only the PI in a mid-term follow-up and PPD at a long-term follow-up reported statistically significant results favoring clear aligners.

Abbreviations: randomized clinical trial, “RCT”; month(s); male, “M”; female, “F”; years old, “y”; number, “n”; month(s), “mo.”; no data found, “NDF”; not specified, “NS”; not statistically significant, “NSS”; statistically significant, “SS”; periodontal probing depth, “PPD”; gingival index, “GI”; plaque index, “PI”; sulcus bleeding index, “SBI”; mean difference, “MD”; standardized mean difference, “SMD”; confidence interval, “CI”; p-value, “P”; fixed appliances, “FA”; clear aligners, “CA”.

**Table 5 healthcare-11-01340-t005:** Methodological quality assessment based on the AMSTAR 2 items.

AMSTAR 2 ITEMS	Jiang et al., 2018 [28]	Lu et al., 2018 [29]	Wu et al., 2022 [30]	Crego-Ruiz et al., 2023 [31]
1. Did the research questions and inclusion criteria for the review include the components of PICO?	Yes	Yes	Yes	Yes
2. Did the report of the review contain an explicit statement that the review methods were established prior to the conduct of the review and did the report justify any significant deviations from the protocol?	**PY**	**PY**	**PY**	**Yes**
3. Did the review authors explain their selection of the study designs for inclusion in the review?	No	Yes	Yes	Yes
4. Did the review authors use a comprehensive literature search strategy?	**Yes**	**Yes**	**Yes**	**Yes**
5. Did the review authors perform study selection in duplicate?	Yes	Yes	Yes	Yes
6. Did the review authors perform data extraction in duplicate?	Yes	Yes	Yes	Yes
7. Did the review authors provide a list of excluded studies and justify the exclusions?	**Yes**	**No**	**No**	**Yes**
8. Did the review authors describe the included studies in adequate detail?	PY	Yes	Yes	Yes
9. Did the review authors use a satisfactory technique for assessing the risk of bias in individual studies that were included in the review?	**Yes**	**Yes**	**Yes**	**Yes**
10. Did the review authors report on the sources of funding for the studies included in the review?	No	Yes	Yes	Yes
11. If meta-analysis was performed did the review authors use appropriate methods for statistical combination of results?	**Yes**	**Yes**	**Yes**	**Yes**
12. If meta-analysis was performed, did the review authors assess the potential impact of risk of bias in individual studies on the results of the meta-analysis or other evidence synthesis?	Yes	Yes	Yes	Yes
13. Did the review authors account for risk of bias in individual studies when interpreting/discussing the results of the review?	**Yes**	**Yes**	**Yes**	**Yes**
14. Did the review authors provide a satisfactory explanation for, and discussion of, any heterogeneity observed in the results of the review?	Yes	Yes	Yes	Yes
15. If they performed quantitative synthesis did the review authors carry out an adequate investigation of publication bias (small study bias) and discuss its likely impact on the results of the review?	**Yes**	**Yes**	**Yes**	**Yes**
16. Did the review authors report any potential sources of conflict of interest, including any funding they received for conducting the review?	Yes	Yes	Yes	Yes

Abbreviations: “PY”; not applicable, “NA”; Bold: critical items.

**Table 6 healthcare-11-01340-t006:** Level of evidence of systematic reviews with meta-analysis included according to the AMSTAR 2 tool.

Level	Description	Jiang et al., 2018 [28]	Lu et al., 2018 [29]	Wu et al., 2022 [30]	Crego-Ruiz et al., 2023 [31]
High	No or one non-critical weakness: the systematic review provides an accurate and comprehensive summary of the results of the available studies that address the question of interest				✓
Moderate	More than one non-critical weakness: the systematic review has more than one weakness but no critical flaws. It may provide an accurate summary of the results of the available studies that were included in the review	✓			
Low	One critical flaw with or without non-critical weaknesses: the review has a critical flaw and may not provide an accurate and comprehensive summary of the available studies that address the question of interest		✓	✓	
Critically low	More than one critical flaw with or without non-critical weaknesses: the review has more than one critical flaw and should not be relied on to provide an accurate and comprehensive summary of the available studies				

**Table 7 healthcare-11-01340-t007:** (**a**) Synthesis results for the PI based on follow-up. (**b**) Synthesis results for the GI based on follow-up. (**c**) Synthesis results for the PPD based on follow-up. (**d**) Synthesis results for the SBI based on follow-up.

**Included Systematic Review**	**(a) PI**		
	**Short-Term Follow-Up** **(from Baseline to 2–3 mo.)**	**Mid-Term Follow-Up** **(from Baseline to 6–9 mo.)**	**Long-Term Follow-Up** **(from Baseline to 12 mo. or More)**
**Jiang et al., 2018** [28]	**1mo.:** MD, −0.35, 95% CI, −0.57 to −0.14 **3 mo.:** MD, −0.63, 95% CI, −1.22 to −0.04	NSS	NSS
**Lu et al., 2018** [29]	**1 mo.:** SMD, −0.53; 95% CI: −0.89 to −0.18 **3 mo.:** SMD, −0.69, 95% CI, −1.12 to −0.27	**6 mo.:** SMD, −0.91, 95% CI, −1.47 to −0.35	N/A
**Wu et al., 2022** [30]	**3 mo.:** MD, −0.57, 95% CI, −0.98 to −0.16	**6 mo.:** MD, −1.10, 95% CI, −1.60 to −0.61	N/A
**Crego-Ruiz et al., 2023** [31]	NSS	MD, −0.99, 95% CI, −1.94 to −0.03	NSS
	**(b) GI**		
	**Short-Term Follow-Up** **(from Baseline to 2–3 mo.)**	**Mid-Term Follow-Up** **(from Baseline to 6–9 mo.)**	**Long-Term Follow-Up** **(from Baseline to 12 mo. or More)**
**Jiang et al., 2018** [28]	**1 mo.:** MD, −0.24, 95% CI, −0.35 to −0.12 **3 mo.:** MD, −0.37, 95% CI, −0.65 to −0.10	N/A	N/A
**Lu et al., 2018** [29]	NSS	NSS	N/A
**Wu et al., 2022** [30]	NSS	**6 mo.:** SMD, −0.91, 95% CI, −1.47 to −0.35; *p* < 0.05	N/A
**Crego-Ruiz et al., 2023** [31]	NSS	NSS	NSS
	**(c) PPD**		
	**Short-Term Follow-Up** **(from Baseline to 2–3 mo.)**	**Mid-Term Follow-Up** **(from Baseline to 6–9 mo.)**	**Long-Term Follow-Up** **(from Baseline to 12 mo. or More)**
**Jiang et al., 2018** [28]	NSS	NSS	12 mo.: MD, −0.45, 95% CI, −0.41 to −0.10; *p* = 0.001
**Lu et al., 2018** [29]	NSS	NSS	N/A
**Wu et al., 2022** [30]	**3 mo.:** MD, −0.26, 95% CI, −0.52 to −0.01	**6 mo.:** MD, −0.42, 95% CI, −0.83 to −0.01	N/A
**Crego-Ruiz et al., 2023** [31]	NSS	NSS	MD, −0.93, 95% CI, −1.16 to −0.70
	**(d) SBI**		
	**Short-Term Follow-Up** **(from Baseline to 2–3 mo.)**	**Mid-Term Follow-Up** **(from Baseline to 6–9 mo.)**	**Long-Term Follow-Up** **(from Baseline to 12 mo. or More)**
**Jiang et al., 2018** [28]	N/A	N/A	N/A
**Lu et al., 2018** [29]	**1 mo.:** SMD, −0.44; 95% CI: −0.70 to −0.19 **3 mo.:** SMD, −0.49, 95% CI, −0.93 to −0.05	**6 mo.:** SMD, −0.91, 95% CI, −1.47 to −0.35	N/A
**Wu et al., 2022** [30]	N/A	N/A	N/A
**Crego-Ruiz et al., 2023** [31]	N/A	N/A	N/A

Abbreviations: month(s), “mo.”; plaque index, “PI”; mean difference, “MD”; standardized mean difference, “SMD”; confidence interval, “CI”; not statistically significant, “NSS”; gingival index, “GI”; periodontal probing depth, “PPD”; sulcus bleeding index, “SBI”; and not available, “N/A”.

## Data Availability

Data supporting the reported results can be found in the Cochrane Library, Web of Science (Core Collection), Scopus, and MEDLINE/PubMed databases.

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
