# Peer review of "Impact of Clear Aligners versus Fixed Appliances on Periodontal Status of Patients Undergoing Orthodontic Treatment: A Systematic Review of Systematic Reviews"

_healthcare, 2023, doi:10.3390/healthcare11091340_

Round 1

Reviewer 1 Report

This study presents a systematic review of other systematic reviews that compare the use of fixed orthodontic appliances and aligners in periodontal health status by clinical parameters. The subject is relevant, and the article is methodologically well outlined and described. Here are a few points to consider.

- The gray literature search was not described in item 2.2 (search strategy). Was it made? If so, include it in the description.

- I suggest including the level of agreement (kappa index) between researchers for the selection of articles, extraction, and data collection.

- Table 2 was a little confusing for the reader. Data that are included in the same column as the study author's name, such as "Studies included in qualitative synthesis", and "Studies included in quantitative meta-analysis" should be presented in two other columns adjacent to "Included systematic review". This way the information is better presented.

Author Response

Dear Reviewer,

thank you for your comment. The revised manuscript according to your correction and observation (highlighted in yellow) was uploaded.

  •  The gray literature search did not provide any relevant records. The manuscript was updated specifying this detail in item 2.2.
  • The level of agreement (Cohen kappa coefficient) between researchers was calculated and included in results (please see item 3.1 line 149-150; item 3.2 line 181 of the updated manuscript).
  • Table 2 was reorganized for better reading and understanding, and a column to present studied included in qualitative and quantitative analysis was included, as you suggested (please see item 3.2, Table 4 of the updated manuscript). 

Reviewer 2 Report

Thank you for the opportunity to review the manuscript on changes of the periodontal status during orthodontic treatment.

The authors should differences in the quality of 4 systematic reviews.

The number of patients in the analysed studies is high enough to present the percentages of different periodontal complications: Loss of periodontal attachment (PD and recessions) and reduction of keratinized gingiva. This important outcome is still missing.

Author Response

Dear Reviewer,

thank you for your comment. The revised manuscript according to most of your correction and observations (highlighted in blue) was uploaded.

  • Your statement is certainly correct: loss of periodontal attachment, gingival recession and reduction of keratinized gingiva are important outcome missing. However, these outcomes were not recorded in the included SRs. Regarding gingival recession, this outcome was reportedin only one recently published study, included in one of the 4 SRs, with a very limited follow-up. The manuscript was updated specifying this detail. Instead, data about reduction of keratinized gingiva were not available.
  • In our opinion, your amendments make the title less clear. Notably, the phrase “a Systematic review of the current evidence from systematic review” helps readers to better understand the design of the study, that is a systematic review of systematic reviews. Thus, we used the diction “Umbrella review”. 
  • Although most aligners are clear, and thus "clear" may seems unnecessary, these appliances are typically named "clear aligners" in literature. 
  • Not all mouthwashes are CHX-based, and  thus we had to specify it in order to accurately report what the reference article stated.
  • Table 2 was presented in a horizontal way, as you suggested (please see Table 4 of the updated manuscript).

Reviewer 3 Report

Thank you very much for this interesting review focusing on a relevant topic in clinical dentistry.

The paper is well organized and written. Only minor english corrections are needed.

The title is correct and precise.

The Abstract is fine including all relevant information.

Introduction:

Please give a clear hypothesis and if possible null-hypothesis of your aim and investigation. Please add.

Material and Methods:

It might be better to organize the search strategy using a table. This might improve readers understanding.

Furthermore, the inclusion and exclusion criteria could be organized in a table.

Results:

Table 2 is very difficult to read. On many pages there is only the third column displaying any data. I would be helpful to reorganize the table for better reading and understanding. Please try to give the information in a readable way.

Table 5 could be shortened to improve understanding. Please reorganize and improve the table. I recommend to put the table on one page.

Conclusion:

Please explain in a short sentence the reason why your results are practically negligible in a clinical context. The significant better results for clear aligners should be explained also in the conclusion.

Author Response

Dear Reviewer,

thank you for your comment. The revised manuscript according to your correction and observations (highlighted in green) was uploaded.

  • A clear hypothesis was formulated, and the null-hypothesis of our aim and investigation was provided (please see Introduction, line 59-63 of the updated manuscript).
  • Search strategy was organized using a table (please see item 2.2, Table 1 of the updated manuscript).
  • Also the inclusion and exclusion criteria were organized in a table (please see item 2.3, Table 2 of the updated manuscript).
  • Table 2 was reorganized for better reading and understanding (please see item 3.2, Table 4 of the updated manuscript).
  • Table 5 was reorganized for better reading and understanding (please see Table 7a, 7b and 7c of the updated manuscript).

Round 2

Reviewer 2 Report

Thank you for your positive corrections in the manuscipt